# Training and Evaluating Multimodal Word Embeddings with Large-scale Web Annotated Images

**Junhua Mao**[1]     **Jiajing Xu**[2]     **Yushi Jing**[2]     **Alan Yuille**[1,3]
[1]University of California, Los Angeles     [2]Pinterest Inc.     [3]Johns Hopkins University
mjhustc@ucla.edu,  {jiajing,jing}@pinterest.com,  alan.l.yuille@gmail.com

## Abstract

In this paper, we focus on training and evaluating effective word embeddings with both text and visual information. More specifically, we introduce a large-scale dataset with 300 million sentences describing over 40 million images crawled and downloaded from publicly available Pins (i.e. an image with sentence descriptions uploaded by users) on Pinterest [2]. This dataset is more than 200 times larger than MS COCO [22], the standard large-scale image dataset with sentence descriptions. In addition, we construct an evaluation dataset to directly assess the effectiveness of word embeddings in terms of finding semantically similar or related words and phrases. The word/phrase pairs in this evaluation dataset are collected from the click data with millions of users in an image search system, thus contain rich semantic relationships. Based on these datasets, we propose and compare several Recurrent Neural Networks (RNNs) based multimodal (text and image) models. Experiments show that our model benefits from incorporating the visual information into the word embeddings, and a weight sharing strategy is crucial for learning such multimodal embeddings. The project page is: `http://www.stat.ucla.edu/~junhua.mao/multimodal_embedding.html`[1].

## 1   Introduction

Word embeddings are dense vector representations of words with semantic and relational information. In this vector space, semantically related or similar words should be close to each other. A large-scale training dataset with billions of words is crucial to train effective word embedding models. The trained word embeddings are very useful in various tasks and real-world applications that involve searching for semantically similar or related words and phrases.

A large proportion of the state-of-the-art word embedding models are trained on pure text data only. Since one of the most important functions of language is to describe the visual world, we argue that the effective word embeddings should contain rich visual semantics. Previous work has shown that visual information is important for training effective embedding models. However, due to the lack of large training datasets of the same scale as the pure text dataset, the models are either trained on relatively small datasets (e.g. [13]), or the visual contraints are only applied to limited number of pre-defined visual concepts (e.g. [21]). Therefore, such work did not fully explore the potential of visual information in learning word embeddings.

In this paper, we introduce a large-scale dataset with both text descriptions and images, crawled and collected from Pinterest, one of the largest database of annotated web images. On Pinterest, users save web images onto their boards (i.e. image collectors) and supply their descriptions of the images. More descriptions are collected when the same images are saved and commented by other users. Compared to MS COCO (i.e. the image benchmark with sentences descriptions [22]), our dataset is much larger (40 million images with 300 million sentences compared to 0.2 million images and 1 million sentences in the current release of MS COCO) and is at the same scale as the standard pure

text training datasets (e.g. Wikipedia Text Corpus). Some sample images and their descriptions are shown in Figure 1 in Section 3.1. We believe training on this large-scale dataset will lead to richer and better generalized models. We denote this dataset as the *Pinterest40M dataset*.

One challenge for word embeddings learning is how to directly evaluate the quality of the model with respect to the tasks (e.g. the task of finding related or similar words and phrases). State-of-the-art neural language models often use the negative log-likelihood of the predicted words as their training loss, which is not always correlated with the effectiveness of the learned embedding. Current evaluation datasets (e.g. [5, 14, 11]) for word similarity or relatedness contain only less than a thousand word pairs and cannot comprehensively evaluate all the embeddings of the words appearing in the training set.

The challenge of constructing large-scale evaluation datasets is partly due to the difficulty of finding a large number of semantically similar or related word/phrase pairs. In this paper, we utilize user click information collected from Pinterest's image search system to generate millions of these candidate word/phrase pairs. Because user click data are somewhat noisy, we removed inaccurate entries in the dataset by using crowdsourcing human annotations. This led to a final gold standard evaluation dataset consists of 10,674 entries.

Equipped with these datasets, we propose, train and evaluate several Recurrent Neural Network (RNN [10]) based models with input of both text descriptions and images. Some of these models directly minimize the Euclidean distance between the visual features and the word embeddings or RNN states, similar to previous work (e.g. [13, 21]). The best performing model is inspired by recent image captioning models [9, 24, 36], with the additional weight-sharing strategy originally proposed in [23] to learn novel visual concepts. This strategy imposes soft constraints between the visual features and all the related words in the sentences. Our experiments validate the effectiveness and importance of incorporating visual information into the learned word embeddings.

We make three major contributions: Firstly, we constructed a large-scale multimodal dataset with both text descriptions and images, which is at the same scale as the pure text training set. Secondly, we collected and labeled a large-scale evaluation dataset for word and phrase similarity and relatedness evaluation. Finally, we proposed and compared several RNN based models for learning multimodal word embeddings effectively. To facilitate research in this area, we will gradually release the datasets proposed in this paper on our project page.

## 2 Related Work

**Image-Sentence Description Datasets** The image descriptions datasets, such as Flickr8K [15], Flickr30K [37], IAPR-TC12 [12], and MS COCO [22], greatly facilitated the development of models for language and vision tasks such as image captioning. Because it takes lots of resources to label images with sentences descriptions, the scale of these datasets are relatively small (MS COCO, the largest dataset among them, only contains 1 million sentences while our Pinterest40M dataset has 300 million sentences). In addition, the language used to describe images in these datasets is relatively simple (e.g. MS COCO only has around 10,000 unique words appearing at least 3 times while there are 335,323 unique words appearing at least 50 times in Pinterest40M). The Im2Text dataset proposed in [28] adopts a similar data collection process to ours by using 1 million images with 1 million user annotated captions from Flickr. But its scale is still much smaller than our Pinterest40M dataset.

Recently, [34] proposed and released the YFCC100M dataset, which is a large-scale multimedia dataset contains metadata of 100 million Flickr images. It provides rich information about images, such as tags, titles, and locations where they were taken. The users' comments can be obtained by querying the Flickr API. Because of the different functionality and user groups between Flickr and Pinterest, the users' comments of Flickr images are quite different from those of Pinterest (e.g. on Flickr, users tend to comment more on the photography techniques). This dataset provides complementary information to our Pinterest40M dataset.

**Word Similarity-Relatedness Evaluation** The standard benchmarks, such as WordSim-353/WS-Sim [11, 3], MEN [5], and SimLex-999 [14], consist of a couple hundreds of word pairs and their similarity or relatedness scores. The word pairs are composed by asking human subjects to write the first related, or similar, word that comes into their mind when presented with a concept word (e.g. [27, 11]), or by randomly selecting frequent words in large text corpus and manually searching for useful pairs (e.g. [5]). In this work, we are able to collect a large number of word/phrase pairs

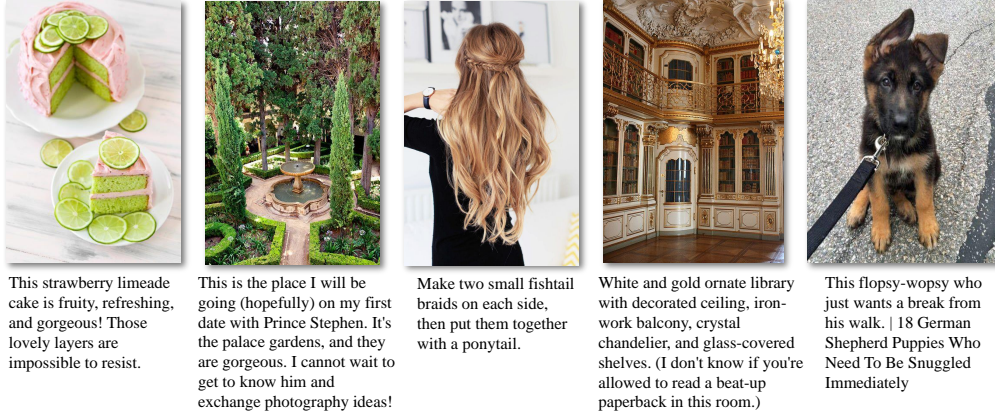

| This strawberry limeade cake is fruity, refreshing, and gorgeous! Those lovely layers are impossible to resist. | This is the place I will be going (hopefully) on my first date with Prince Stephen. It's the palace gardens, and they are gorgeous. I cannot wait to get to know him and exchange photography ideas! | Make two small fishtail braids on each side, then put them together with a ponytail. | White and gold ornate library with decorated ceiling, iron-work balcony, crystal chandelier, and glass-covered shelves. (I don't know if you're allowed to read a beat-up paperback in this room.) | This flopsy-wopsy who just wants a break from his walk. | 18 German Shepherd Puppies Who Need To Be Snuggled Immediately |

Figure 1: Sample images and their sample descriptions collected from Pinterest.

with good quality by mining them from the click data of Pinterest's image search system used by millions of users. In addition, because this dataset is collected through a visual search system, it is more suitable to evaluate multimodal embedding models. Another related evaluation is the analogy task proposed in [25]. They ask the model questions like "man to woman is equal king to what?" as their evaluation. But such questions do not directly measure the word similarity or relatedness, and cannot cover all the semantic relationships of million of words in the dictionary.

**RNN for Language and Vision** Our models are inspired by recent RNN-CNN based image captioning models [9, 24, 36, 16, 6, 18, 23], which can be viewed as a special case of the sequence-to-sequence learning framework [33, 7]. We adopt Gated Recurrent Units (GRUs [7]), a variation of the simple RNN model.

**Multimodal Word Embedding Models** For pure text, one of the most effective approaches to learn word embeddings is to train neural network models to predict a word given its context words in a sentence (i.e. the continuous bag-of-word model [4]) or to predict the context words given the current word (i.e. the skip-gram model [25]). There is a large literature on word embedding models that utilize visual information. One type of methods takes a two-step strategy that first extracts text and image features separately and then fuses them together using singular value decomposition [5], stacked autoencoders [31], or even simple concatenation [17]. [13, 21, 19] learn the text and image features jointly by fusing visual or perceptual information in a skip-gram model [25]. However, because of the lack of large-scale multimodal datasets, they only associate visual content with a pre-defined set of nouns (e.g. [21]) or perception domains (e.g. [14]) in the sentences, or focus on abstract scenes (e.g. [19]). By contrast, our best performing model places a soft constraint between visual features and all the words in the sentences by a weight sharing strategy as shown in Section 4.

## 3 Datasets

We constructed two datasets: one for training our multimodal word-embeddings (see Section 3.1) and another one for the evaluation of the learned word-embeddings (see Section 3.2).

### 3.1 Training Dataset

Pinterest is one of the largest repository of Web images. Users commonly tag images with short descriptions and share the images (and desriptions) with others. Since a given image can be shared and tagged by *multiple, sometimes thousands* of users, many images have a very rich set of descriptions, making this source of data ideal for training model with both text and image inputs.

Table 1: Scale comparison with other image descriptions benchmarks.

|  | Image | Sentences |
|---|---|---|
| Flickr8K [15] | 8K | 40K |
| Flickr30K [37] | 30K | 150K |
| IAPR-TC12 [12] | 20K | 34K |
| MS COCO [22] | 200K | 1M |
| Im2Text [28] | 1M | 1M |
| Pinterset40M | 40M | 300M |

The dataset is prepared in the following way: first, we crawled the public available data on Pinterest to construct our training dataset of more than 40 million images. Each image is associated with an average of 12 sentences, and we removed duplicated or short sentences with less than 4 words. The duplication detection is conducted by calculating the

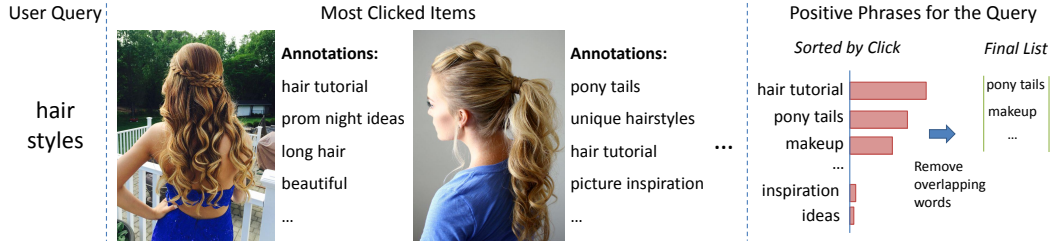

Figure 2: The illustration of the positive word/phrase pairs generation. We calculate a score for each annotation (i.e. a short phrase describes the items) by aggregating the click frequency of the items to which it belongs and rank them according to the score. The final list of positive phrases are generated from the top ranked phrases after removing phrases containing overlapping words with the user query phrase. See text for details.

overlapped word unigram ratios. Some sample images and descriptions are shown in Figure 1. We denote this dataset as the *Pinterest40M* dataset.

Our dataset contains 40 million images with 300 million sentences (around 3 billion words), which is much larger than the previous image description datasets (see Table 1). In addition, because the descriptions are annotated by users who expressed interest in the images, the descriptions in our dataset are more natural and richer than the annotated image description datasets. In our dataset, there are 335,323 unique words with a minimum number of occurence of 50, compared with 10,232 and 65,552 words appearing at least 3 times in MS COCO and IM2Text dataset respectively. To the best of our knowledge, there is no previous paper that trains a multimodal RNN model on a dataset of such scale.

## 3.2 Evaluation Datasets

This work proposes to use labeled phrase triplets – each triplet is a three-phrase tuple containing phrase A, phrase B and phrase C, where A is considered as semantically closer to B than A is to C. At testing time, we compute the distance in the word embedding space between A/B and A/C, and consider a test triplet as positive if $d(A, B) < d(A, C)$. This relative comparison approach was commonly used to evaluate and compare different word embedding models [30].

In order to generate large number of phrase triplets, we rely on user-click data collected from Pinterest image search system. At the end, we construct a large-scale evaluation dataset with 9.8 million triplets (see Section 3.2.1), and its cleaned up gold standard version with 10 thousand triplets (see Section 3.2.2).

### 3.2.1 The Raw Evaluation Dataset from User Clickthrough Data

It is very hard to obtain a large number of semantically similar or related word and phrase pairs. This is one of the challenges for constructing a large-scale word/phrase similarity and relatedness evaluation dataset. We address this challenge by utilizing the user clickthrough data from Pinterest image search system, see Figure 2 for an illustration.

More specifically, given a query from a user (e.g. "hair styles"), the search system returns a list of items, and each item is composed of an image and a set of annotations (i.e. short phrases or words that describe the item). Please note that the same annotation can appear in multiple items, e.g., "hair tutorial" can describe items related to prom hair styles or ponytails. We derive a matching score for each annotation by aggregating the click frequency of the items containing the annotation. The annotations are then ranked according to the matching scores, and the top ranked annotations are considered as the positive set of phrases or words with respect to the user query.

To increase the difficulty of this dataset, we remove the phrases that share common words with the user query from the initial list of positive phrases. E.g. "hair tutorials" will be removed because the word "hair" is contained in the query phrase "hair styles". A stemmer in Python's "stemmer" package is also adopted to find words with the same root (e.g. "cake" and "cakes" are considered as the same word). This pruning step also prevents giving bias to methods which measure the similarity between the positive phrase and the query phrase by counting the number of overlapping words between them. In this way, we collected 9,778,508 semantically similar phrase pairs.

Table 2: Sample triplets from the Gold RP10K dataset.

| Base Phrase | Positive Phrase | Negative Phrase |
|---|---|---|
| hair style | ponytail | pink nail |
| summer lunch | salads sides | packaging bottle |
| oil painting ideas | art tips | snickerdoodle muffins |
| la multi ani birthdays | wishes | tandoori |
| teach activities | preschool | rental house ideas |
| karting | go carts | office waiting area |
| looking down | the view | soft curls for medium hair |
| black ceiling | home ideas | paleo potluck |
| new marriage quotes | true love | winter travel packing |
| sexy scientist costume | labs | personal word wall |
| framing a mirror | decorating bathroom | celebrity style inspiration |

Previous word similarity/relatedness datasets (e.g. [11, 14]) manually annotated each word pair with an absolute score reflecting how much the words in this pair are semantically related. In the testing stage, a predicted similarity score list of the word pairs generated by the model in the dataset is compared with the groundtruth score list. The Spearman's rank correlation between the two lists is calculated as the score of the model. However, it is often too hard and expensive to label the absolute related score and maintain the consistency across all the pairs in a large-scale dataset, even if we average the scores of several annotators.

We adopt a simple strategy by composing triplets for the phrase pairs. More specifically, we randomly sample negative phrases from a pool of 1 billion phrases. The negative phrase should not contain any overlapping word (a stemmer is also adopted) with both of the phrases in the original phrase pair. In this way, we construct 9,778,508 triplets with the format of (base phrase, positive phrase, negative phrase). In the evaluation, a model should be able to distinguish the positive phrase from the negative phrase by calculating their similarities with the base phrase in the embedding space. We denote this dataset as *Related Phrase 10M (RP10M)* dataset.

### 3.2.2 The Cleaned-up Gold Standard Dataset

Because the raw Related Query 10M dataset is built upon user click information, it contains some noisy triplets (e.g. the positive and base phrase are not related, or the negative phrase is strongly related to the base phrase). To create a gold standard dataset, we conduct a clean up step using the crowdsourcing platform CrowdFlower [1] to remove these inaccurate triplets. A sample question and choices for the crowdsourcing annotators are shown in Figure 3. The positive and negative phrases in a triplet are randomly given as choice "A" or "B". The annotators are required to choose which phrase is more related to the base phrase, or if they are both related or unrelated. To help the annotators understand the meaning of the phrases, they can click on the phrases to get Google search results.

We annotate 21,000 triplets randomly sampled from the raw Related Query 10M dataset. Three to five annotators are assigned to each question. A triplet is accepted and added in the final cleaned up dataset only if more than 50% of the annotators agree with the original positive and negative label of the queries (note that they do not know which one is positive in the annotation process). In practice, 70% of the selected phrases triplets have more than 3 annotators to agree. This leads to a gold standard dataset with 10,674 triplets. We denote this dataset as *Gold Phrase Query 10K (Gold RP10K)* dataset.

Figure 3: The interface for the annotators. They are required to choose which phrase (positive and negative phrases will be randomly labeled as "A" or "B") is more related to base phrase. They can click on the phrases to see Google search results.

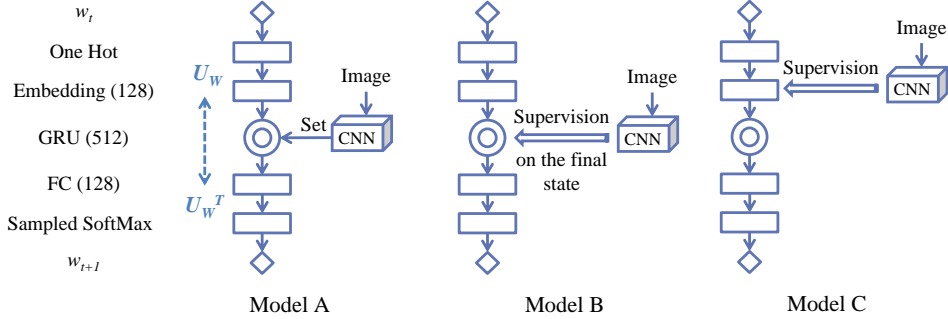

Figure 4: The illustration of the structures of our model A, B, and C. We use a CNN to extract visual representations and use a RNN to model sentences. The numbers on the bottom right corner of the layers indicate their dimensions. We use a sampled softmax layer with 1024 negative words to accelerate the training. Model A, B, and C differ from each other by the way that we fuse the visual representation into the RNN. See text for more details.

This dataset is very challenging and a successfully model should be able to capture a variety of semantic relationships between words or phrases. Some sample triplets are shown in Table 2.

## 4 The Multimodal Word Embedding Models

We propose three RNN-CNN based models to learn the multimodal word embeddings, as illustrated in Figure 4. All of the models have two parts in common: a Convolutional Neural Network (CNN [20]) to extract visual representations and a Recurrent Neural Network (RNN [10]) to model sentences.

For the CNN part, we resize the images to $224 \times 224$, and adopt the 16-layer VGGNet [32] as the visual feature extractor. The binarized activation (i.e. 4096 binary vectors) of the layer before its SoftMax layer are used as the image features and will be mapped to the same space of the state of RNN (Model A, B) or the word embeddings (Model C), depends on the structure of the model, by a fully connected layer and a Rectified Linear Unit function (ReLU [26], $\text{ReLU}(x) = \max(0, x)$).

For the RNN part, we use a Gated Recurrent Unit (GRU [7]), an recently very popular RNN structure, with a 512 dimensional state cell. The state of GRU $h_t$ for each word with index $t$ in a sentence can be represented as:

$$r_t = \sigma(W_r[e_t, h_{t-1}] + b_r) \tag{1}$$

$$u_t = \sigma(W_u[e_t, h_{t-1}] + b_u) \tag{2}$$

$$c_t = \tanh(W_c[e_t, r_t \odot h_{t-1}] + b_c) \tag{3}$$

$$h_t = u_t \odot h_{t-1} + (1 - u_t) \odot c_t \tag{4}$$

where $\odot$ represents the element-wise product, $\sigma(.)$ is the sigmoid function, $e_t$ denotes the word embedding for the word $w_t$, $r_t$ and $u_t$ are the reset gate and update gate respectively. The inputs of the GRU are words in a sentence and it is trained to predict the next words given the previous words.

We add all the words that appear more than 50 times in the Pinterest40M dataset into the dictionary. The final vocabulary size is 335,323. Because the vocabulary size is very huge, we adopt the sampled SoftMax loss [8] to accelerate the training. For each training step, we sample 1024 negative words according to their log frequency in the training data and calculate the sampled SoftMax loss for the positive word. This sampled SoftMax loss function of the RNN part is adopted with Model A, B and C. Minimizing this loss function can be considered as approximately maximizing the probability of the sentences in the training set.

As illustrated in Figure 4, Model A, B and C have different ways to fuse the visual information in the word embeddings. Model A is inspired by the CNN-RNN based image captioning models [36, 23]. We map the visual representation in the same space as the GRU states to initialize them (i.e. set $h_0 = \text{ReLU}(W_I f_I)$). Since the visual information is fed after the embedding layer, it is usually hard to ensure that this information is fused in the learned embeddings. We adopt a transposed weight sharing strategy proposed in [23] that was originally used to enhance the models' ability to learn novel visual concepts. More specifically, we share the weight matrix of the SoftMax layer $U_M$ with the matrix $U_w$ of the word embedding layer in a transposed manner. In this way, $U_w^T$ is learned to decode the visual information and is enforced to incorporate this information into the word embedding matrix

Table 3: Performance comparison of our Model A, B, C, their variants and a state-of-the-art skip-gram model [25] trained on Google News dataset with 300 billion words.

| | Gold RP10K | RP10M | dim |
|---|---|---|---|
| Pure text RNN | 0.748 | 0.633 | 128 |
| Model A without weight sharing | 0.773 | 0.681 | 128 |
| Model A (weight shared multimodal RNN) | **0.843** | **0.725** | 128 |
| Model B (direct visual supervisions on the final RNN state) | 0.705 | 0.646 | 128 |
| Model C (direct visual supervisions on the embeddings) | 0.771 | 0.687 | 128 |
| Word2Vec-GoogleNews [25] | 0.716 | 0.596 | 300 |
| GloVe-Twitter [29] | 0.693 | 0.617 | 200 |

$U_w$. In the experiments, we show that this strategy significantly improve the performance of the trained embeddings. Model A is trained by maximizing the log likelihood of the next words given the previous words conditioned on the visual representations, similar to the image captioning models.

Compared to Model A, we adopt a more direct way to utilize the visual information for Model B and Model C. We add direct supervisions of the final state of the GRU (Model B) or the word embeddings (Model C), by adding new loss terms, in addition to the negative log-likelihood loss from the sampled SoftMax layer:

$$\mathcal{L}_{state} = \frac{1}{n} \sum_s \parallel h_{l_s} - \text{ReLU}(W_I f_{I_s}) \parallel \tag{5}$$

$$\mathcal{L}_{emb} = \frac{1}{n} \sum_s \frac{1}{l_s} \sum_t \parallel e_t - \text{ReLU}(W_I f_{I_s}) \parallel \tag{6}$$

where $l_s$ is the length of the sentence $s$ in a mini-batch with $n$ sentences, Eqn. 5 and Eqn. 6 denote the additional losses for model B and C respectively. The added loss term is balanced by a weight hyperparameter $\lambda$ with the negative log-likelihood loss from the sampled SoftMax layer.

## 5 Experiments

### 5.1 Training Details

We convert the words in all sentences of the Pinterest40M dataset to lower cases. All the non-alphanumeric characters are removed. A start sign $\langle bos \rangle$ and an end sign $\langle eos \rangle$ are added at the beginning and the end of all the sentences respectively.

We use the stochastic gradient descent method with a mini-batch size of 256 sentences and a learning rate of 1.0. The gradient is clipped to 10.0. We train the models until the loss does not decrease on a small validation set with 10,000 images and their descriptions. The models will scan the dataset for roughly five 5 epochs. The bias terms of the gates (i.e. $b_r$ and $b_u$ in Eqn. 1 and 2) in the GRU layer are initialized to 1.0.

### 5.2 Evaluation Details

We use the trained embedding models to extract embeddings for all the words in a phrase and aggregate them by average pooling to get the phrase representation. We then check whether the cosine distance between the (base phrase, positive phrase) pair are smaller than the (base phrase, negative phrase) pair. The average precision over all the triplets in the raw Related Phrases 10M (RP10M) dataset and the Gold standard Related Phrases 10K (Gold RP10K) dataset are reported.

### 5.3 Results on the Gold RP10K and RP10M datasets

We evaluate and compare our Model A, B, C, their variants and several strong baselines on our RP10M and Gold RP10K datasets. The results are shown in Table 3. "Pure Text RNN" denotes the baseline model without input of the visual features trained on Pinterest40M. It have the same model structure as our Model A except that we initialize the hidden state of GRU with a zero vector. "Model A without weight sharing" denotes a variant of Model A where the weight matrix $U_w$ of the word embedding layer is not shared with the weight matrix $U_M$ of the sampled SoftMax layer (see Figure 4 for details). [2] "Word2Vec-GoogleNews" denotes the state-of-the-art off-the-shelf word

embedding models of Word2Vec [25] trained on the Google-News data (about 300 billion words). "GloVe-Twitter" denotes the GloVe model [29] trained on the Twitter data (about 27 billion words). They are pure text models, but trained on a very large dataset (our model only trains on 3 billion words). Comparing these models, we can draw the following conclusions:

- Under our evaluation criteria, visual information significantly helps the learning of word embeddings when the model successfully fuses the visual and text information together. E.g., our Model A outperforms the Word2Vec model by 9.5% and 9.2% on the Gold RP10K and RP10M datasets respectively. Model C also outperforms the pure text RNN baselines.

- The weight sharing strategy is crucial to enhance the ability of Model A to fuse visual information into the learned embeddings. E.g., our Model A outperforms the baseline without this sharing strategy by 7.0% and 4.4% on Gold RP10K and RP10M respectively.

- Model A performs the best among all the three models. It shows that soft supervision imposed by the weight-sharing strategy is more effective than direct supervision. This is not surprising since not all the words are semantically related to the content of the image and a direct and hard constraint might hinder the learning of the embeddings for these words.

- Model B does not perform very well. The reason might be that most of the sentences have more than 8 words and the gradient from the final state loss term $\mathcal{L}_{state}$ cannot be easily passed to the embedding of all the words in the sentence.

- All the models trained on the Pinterest40M dataset performs better than the skip-gram model [25] trained on a much larger dataset of 300 billion words.

## 6 Discussion

In this paper, we investigate the task of training and evaluating word embedding models. We introduce *Pinterest40M*, the largest image dataset with sentence descriptions to the best of our knowledge, and construct two evaluation dataset (i.e. *RP10M* and *Gold RP10K*) for word/phrase similarity and relatedness evaluation. Based on these datasets, we propose several CNN-RNN based multimodal models to learn effective word embeddings. Experiments show that visual information significantly helps the training of word embeddings, and our proposed model successfully incorporates such information into the learned embeddings.

There are lots of possible extensions of the proposed model and the dataset. E.g., we plan to separate semantically similar or related phrase pairs from the Gold RP10K dataset to better understand the performance of the methods, similar to [3]. We will also give relatedness or similarity scores for the pairs (base phrase, positive phrase) to enable same evaluation strategy as previous datasets (e.g. [5, 11]). Finally, we plan to propose better models for phrase representations.

**Acknowledgement** We are grateful to James Rubinstein for setting up the crowdsourcing experiments for dataset cleanup. We thank Veronica Mapes, Pawel Garbacki, and Leon Wong for discussions and support. We appreciate the comments and suggestions from anonymous reviewers of NIPS 2016. This work is partly supported by the Center for Brains, Minds and Machines NSF STC award CCF-1231216 and the Army Research Office ARO 62250-CS.

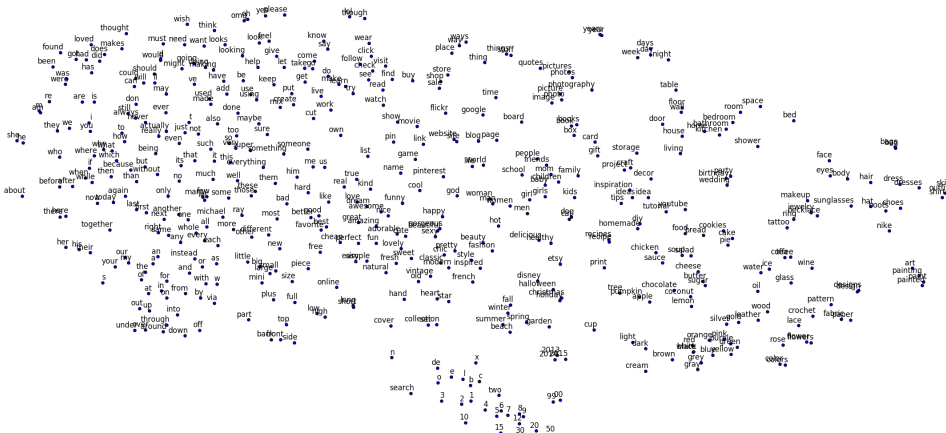

Figure 5: t-SNE [35] visualization of the 500 most frequent words learned by our Model A.

## Footnotes

[1]The datasets introduced in this work will be gradually released on the project page.

[2]We also try to adopt the weight sharing strategy in Model B and C, but the performance is very similar to the non-weight sharing version.

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
