[Reviews · NeurIPS 2016]

Reviewer 1

Summary

This paper examines training and evaluating multi-modal word embeddings with large data sets. It contributes a new data set of images with captions derived from Pinterest, and a new data set of phrase similarity judgments automatically derived from click through data. A subset of these judgments were checked in a user study. A number of RNN-CNN models are trained on the Pinterest data, and evaluated on the newly create data sets. It is found that a multi-modal RNN with a transposed weight sharing scheme between the input word embeddings and the output layer fed to the softmax word prediction layer achieves the best performance on the evaluations.

Qualitative Assessment

The paper was clear and well written. The data set and the evaluation that was conducted could be useful to the community. However, the paper unfairly characterizes or omits some previous work, and was not clear enough about the limitations and biases of their evaluation strategy. These points detract from a paper that otherwise makes an interesting contribution. First, there is an implied criticism of WordSim-353 and MEN at the bottom of page 2 that they only contain similarity judgments at the word level. However, there is a large amount of work on learning phrase and sentence-level embeddings in the recently literature that overcome these issues (see representative work by Mirella Lapata, Marco Baroni, Stephen Clarke, Richard Socher, among many others), which the paper does not mention. Thus, learning 2- or 3-word embeddings is already well investigated, rather than a source of new challenges. The criticism of the analogical reasoning task on page 3 is also misplaced. The paper criticizes this task for not covering all the semantic relationships of millions of words in the dictionary. In my view, relatedness judgments are much worse than analogical reasoning, because they reduce all semantic relations down to a single scalar. The paper should be up front about its limitations and biases. The data collection process the resulting evaluation are clearly biased towards multi-modal methods, because an image is displayed in the interface along with the text. This is not a problem, but the fact that multi-modal representations outperform pure text ones is then less meaningful, and by no means spells the end of models trained purely on text. Also, the paper should discuss possible confounds with the construction of the click-through evaluation data. A link can be clicked on for reasons other than relevance or similarity between query and the phrase that is presented. It seems that other factors are involved as well (e.g., catchiness of the phrase or the perceived informativeness of the linked article). The paper would be stronger if the multi-modal methods were evaluated on the WS 353 and MEN data sets as well. This would give an indication of whether they might outperform pure text models on tasks that were originally conceived for text models, at least on words related to visual imagery. Finally, the decision to omit cases involving word overlap is understandable, but it does not come without a cost. Accounting for homonymy or polysemy would not be tested by their approach.

Confidence in this Review

2-Confident (read it all; understood it all reasonably well)


Reviewer 2

Summary

This paper presents a new multi-million image-caption style dataset and use it for training multimodal embedding models. They also present a new user click based similarity dataset for evaluation. Finally, they try 2-3 CNN-RNN style models for multimodal phrase embedding similarity. The datasets are decent contributions (but with some issues). However, the evaluation has multiple issues in its current form and the models are also borrowed from previous work.

Qualitative Assessment

After author response: Thanks for answering some of my questions -- I updated some of my scores accordingly. I still encourage the authors to answer the rest of the questions, esp. eval on better downstream tasks like captioning and visual zero-shot learning. --------------------- Evaluation issues: -- No existing datasets have been used to evaluate, e.g., WordSim353, SimLex-999 or the new visual similarity datasets VisSim and SemSim from Lazaridou et al., 2015. The authors have not compared to any existing paper/model on these datasets to show the advantage of either their larger/better training dataset or their models, hence leaving no takeaways. -- Why only work on word similarity? Why not show other image-language tasks in addition, e.g., captioning, visual zero-shot learning? This will better demonstrate the advantages of the training dataset. -- For word embedding baseline, please use stronger latest models such as GloVe and Skip-thought and paragram. Evaluation dataset issues: -- The dataset mixes multiple types of similarity, e.g., related versus paraphrase/synonym, and this has been discussed to be a big issue for evaluating embedding models. -- the data uses triplets instead of the standard ranking+correlation method, and then they also choose random negative phrases for these triplets (which will be easy to detect), which makes the task much easier. -- In the CrowdFlower experiment, the turkers will assume they need to choose at least one of the two phrases most of the time, and even if they choose the correct phrase, it might be only because it is more related to the query phrase as compared to the other random negative phrase, but overall the positive phrase might still be only very slightly related to the query phrase (because of the initial recommendation system based retrieval). To verify this, the authors should get turkers to also rate how much these 'positive' phrases are related to the query phrase in absolute terms/ratings. -- they use 3-5 annotators and then choose phrases where >50% of the annotators agree, but this will mean agreement between just two people in many cases. For such a noisy initial dataset, the filtering should be more strict than 50%, or the #annotators should be higher. Training dataset: -- Since the data is from Pinterest and the 'captions' are just user comments, I am worried that some of these captions might not be standard description style sentences but instead might contain some non-image/visual story or information -- the authors should investigate and present this, and also verify if this corpus can be used for training captioning systems. Model: -- main model figure is very unclear and should be expanded and maybe separated. -- all models are from previous work; would have been good to also suggest some new model variants for this task/datasets. -- why use average pooling at test time for a phrase representation and not run the trained RNN model? Other: -- Lots of typo's throughout the paper, e.g., Line15: "crutial" --> "crucial" Line34: "commemted" --> "commented" Table 3: "weigh" --> "weight" Line 232: "to Model A, We" --> "to Model A, we"

Confidence in this Review

3-Expert (read the paper in detail, know the area, quite certain of my opinion)


Reviewer 3

Summary

This paper introduces a wonderful new image-sentence dataset. It should be a great resource for multimodal research and training. I hope it will come with a good license. The model isn't that interesting and NIPS cares historically more about that and hence misses out on some impactful dataset papers.

Qualitative Assessment

I'd like to see some more analysis of the dataset. number and distribution of unique words. problems like personal comments (vs visual descriptions) and ungrammaticality etc.

Confidence in this Review

3-Expert (read the paper in detail, know the area, quite certain of my opinion)


Reviewer 4

Summary

This paper introduces a new large scale dataset of annotated images from the web. More precisely, the authors crawled approximately 40 millions images from the website interest, along with descriptions submitted by users. There is an average of 12 sentences per image, and many images are described by multiple users. The authors also introduce a new dataset for evaluating word representations. This dataset is made of triplets of short phrase, first two phrases being semantically closer than the first phrase and the third phrase. The positive phrase pairs were obtained using click data, while the negative pairs were randomly sampled. This dataset contains approximately 9.8 millions triplets. The authors also manually cleaned 10,000 triplets, using a crowdsourcing platform. Finally, the authors propose different baselines to learn word vector representations using visual information, based on this dataset. More precisely, they describe three RNN-CNN models, inspired by models for caption generation. They apply these models on the proposed dataset, showing that using multi-modal data is helpful for this evaluation dataset. In particular, they show that on the proposed evaluation set, the proposed models outperforms the pre-trained word2vec vectors (which were trained on approx. 300 billion words).

Qualitative Assessment

This paper is very clearly written. It introduces two large scale datasets, which could have a big impact for researchers working on learning models from multi-modal data. I believe that collecting and sharing high quality dataset is important for the machine learning community, and it seems to me that the Pinterest 40M images could be such a dataset. However, I have a couple of concerns regarding this paper. First, the paper does not mention the Yahoo Flickr Creative Commons (YFCC) dataset, which contains approximately 100 millions images from Yahoo Flickr. This dataset also contains images with descriptions provided by users. It also contains other metadata, such as tags, location or time. While I believe the two datasets are different, I think the authors should discuss the difference between the two (and not claim that they propose a dataset "200 times larger than the current multimodal datasets"). Second, I think that baselines simpler than RNN-CNN should be considered in the paper. Examples of such baselines are: - use the skipgram or cbow models from word2vec on the descriptions (pure text baseline) ; - use the multimodal skipgram described in [Lazaridou et al.]. Overall, I enjoyed reading this paper and I am looking forward the release of this dataset. However, I believe that this paper would be stronger with better discussion of existing datasets and baselines for multimodal data. == Additional comments == A classical evaluation methodology in for multimodal data is retrieval: given a description, is the model able to retrieve the corresponding image. Have the authors considered this tasks? [Thomee et al.] YFCC100M: The New Data in Multimedia Research (http://webscope.sandbox.yahoo.com/catalog.php?datatype=i&did=67) [Lazaridou et al.] Combining Language and Vision with a Multimodal Skip-gram Model

Confidence in this Review

2-Confident (read it all; understood it all reasonably well)


Reviewer 5

Summary

The paper provides a method to fuse the visual information in the word embeddings and it tries to prove that the visual information is able to improve the performance of the word embeddings by semantical similarity.

Qualitative Assessment

1. The conclusion in line 281-282 which is not fair, since T.Mikolov's method and the author's model are trained by different datasets. It is not able to see if the performance is improved by the author's model or just the dataset. Is it possible to train the T.Mikolov's model with your dataset? 2. Could you describe a little more about your baseline (Model A without visual) in line 259-260. Otherwise, it is not clear to see if the performance improvement comes from the visual information of the image or just the relationship between the descriptions and images.

Confidence in this Review

2-Confident (read it all; understood it all reasonably well)


Reviewer 6

Summary

Pinterest is crawled to generate a sentence/image aligned corpus of 300M sentences/40M images. This data is used to train joint image-language models in the spirit of image caption training, with the intent of learning word embeddings with visual information. Another corpus is collected which contains 10M semantically similar phrases of the form base phrase, positive phrase, negative phrase. 22k randomly selected samples from this corpus is cleaned up resulting in 10k triplets used for evaluation of the multi-modal trained word embeddings, given three phrases 1,2,3 the distance in word embedding space, d(1,2) and d(1,3) where it is known 1 is semantically closer to 2 than 3, system must declare d(1,2) < d(1,3) to be correct. Three multi-modal trained word embedding models are investigated, with all 3 using VGG based image features, and GRU to model the language. Two models augment the base loss of maximizing the sentence probability with MSE losses from (1) embedded GRU hidden state to embedded image and (2) embedded words to embedded image. The best model uses only the base loss of sentence probability and a weight sharing strategy between word embedding and softmax output. This model shows a strong advantage to using multimodal trained word embeddings vs. text only trained word embeddings.

Qualitative Assessment

The collection of multi-modal data from a new source Pinterest and the large semantic relatedness dataset is very important to the community. Comparisons are made based on size of corpus to MSCOCO/Flickr30,8 and SBU Im2Text, but no discussion on quality of data in the comparison. MSCOCO and Flickr data are carefully labelled for tight alignment with what is in the image and the sentences, whereas Im2Text is not, using the Flickr sentences rawly from the people who posted the image. Because of this we may train and evaluate caption and bidirectional retrieval systems reliably on MSCOCO and Flickr, but this is not the case with Im2text, often what is in that data are sentences not talking about the image. The sample images of figure 1 in the paper show that Pinterest sentences may include a lot of information outside of the context the sentence, so it is not clear how generally useful this new set is. Certainly you can train better word embeddings using the visual information to measure semantic relatedness, so there is some alignment, but this might be more forgiving than the difficult tasks of captioning and multi-modal retrieval? More should be said about the image extraction pipeline, since VGG is used at level before softmax, you must have used a single crop of multiple crops and averaged? Also the inclusion in model B and C of MSE objectives, this seems reasonable forcing sentence embeddings either from hidden state or word embeddings to match embedded image, but it is not working, this probably requires more investigation, and also more details of the RNN/GRU used. Was this a single layer or two layer GRU? If single does it make sense that the resulting sentence embedding being responsible for generating the words should also be able to create a sentence embedding that converges to the sentence embedding.

Confidence in this Review

2-Confident (read it all; understood it all reasonably well)